# Safeguarding Blockchain Ecosystem: Understanding and Detecting Attack Transactions on Cross-chain Bridges

Submission Id: 1394

## Abstract

Cross-chain bridges are essential decentralized applications (DApps) to facilitate interoperability between different blockchain networks. Unlike regular DApps, the functionality of cross-chain bridges relies on the collaboration of information both on and off the chain, which exposes them to a wider risk of attacks. According to our statistics, attacks on cross-chain bridges have resulted in losses of nearly 4.3 billion dollars since 2021. Therefore, it is particularly necessary to understand and detect attacks on cross-chain bridges. In this paper, we collect the largest number of cross-chain bridge attack incidents to date, including 49 attacks that occurred between June 2021 and September 2024. Our analysis reveal that attacks against cross-chain business logic cause significantly more damage than those that do not. These cross-chain attacks exhibit different patterns compared to normal transactions in terms of call structure, which effectively indicates potential attack behaviors. Given the significant losses in these cases and the scarcity of related research, this paper aims to detect attacks against cross-chain business logic, and propose the BridgeGuard tool. Specifically, BridgeGuard models cross-chain transactions from a graph perspective, and employs a two-stage detection framework comprising global and local graph mining to identify attack patterns in cross-chain transactions. We conduct multiple experiments on the datasets with 203 attack transactions and 40,000 normal cross-chain transactions. The results show that BridgeGuard's reported recall score is 36.32% higher than that of state-of-the-art tools and can detect unknown attack transactions.

## CCS Concepts

• **Security and privacy → Web application security**; • **Applied computing → Electronic funds transfer**.

## Keywords

Blockchain, Cross-Chain, Transaction Analysis, Graph mining

**ACM Reference Format:**

Anonymous Author(s). 2025. Safeguarding Blockchain Ecosystem: Understanding and Detecting Attack Transactions on Cross-chain Bridges. In *Proceedings of The ACM Web Conference (WWW '25)*. ACM, New York, NY, USA, 11 pages. https://doi.org/XXXXXXX.XXXXXXX

## 1 Introduction

In blockchain technology, each blockchain network constitutes a relatively independent ecosystem with its own rules, protocols, and characteristics. This isolation leads to the mutual isolation between blockchains, where even the same cryptocurrency can only be used on specific blockchain networks. For example, Ether is the native cryptocurrency on the Ethereum network [37]. If someone needs to use Ether on other blockchains, they typically have to undergo a series of complex exchange transactions, which inconvenience users and impose limitations. Therefore, with the rapid development of current blockchain technology and the formation of a multi-chain ecosystem, cross-chain bridges, as decentralized applications (DApps), have emerged to bridge this gap, providing users with solutions to achieve interoperability of assets between different blockchain networks.

Cross-chain bridges, through smart contracts and other technical means, enables users to swiftly transfer assets between different blockchain networks, thus achieving cross-chain liquidity of assets. According to DappRadar[1], there are currently over 440 cross-chain bridge DApps implemented based on various cross-chain mechanisms, making them an indispensable part of the blockchain ecosystem. For example, Celer cBridge operates based on the Hash Time Lock Contract (HTLC) mechanism [41], Poly Network operates based on Relay Chain [30], and MultiChain operates based on Notary Mechanism [39]. With the increasing number of blockchain networks, the importance and demand for cross-chain bridges are gradually becoming more prominent.

However, as bridges between different blockchain networks, cross-chain bridges often carry a substantial amount of asset value, and their hidden vulnerabilities make them targets for hackers. In recent years, many security incidents of cross-chain bridges have emerged. The top three security incidents with the highest losses in the Rekt attack database[2] are all related to cross-chain bridges, with Ronin Network losing $624 million, Poly Network losing $611 million, and BNB Bridge losing $586 million, respectively. Particularly, Thorchain suffered three attacks within just two months (on June 29, July 16, and July 23, 2021). The frequency of these security incidents indicates that the security issues of cross-chain bridges have become a significant challenge in the current blockchain field.

Despite the attention paid to the safety of cross-chain bridges, relatively few studies and solutions have been developed in this area. Although some studies have explored and analyzed the safety issues of cross-chain bridges, there are still some limitations. For example, a recent work Xscope [42] focuses on investigating and summarising the security incidents of cross-link bridges occurring from 2021 to March 2022 and gives a rule-based detection method. And some

---

[1]https://dappradar.com/rankings/defi/18?category=defi_cross-chain
[2]https://rekt.news/zh/leaderboard/

*Systematisation of Knowledge Papers* (SoK) studies [15, 16, 21] classify and discuss the attack surface, defense methods and problems of cross-chain bridges. However, as cross-chain mechanisms are still under development and refinement, and there are various attacks against cross-chain bridges. There is still a lack of comprehensive analyses of cross-chain bridge attacks as well as scalable solutions.

**Scope and Contributions.** In order to provide valuable insights for enhancing cross-chain bridge security, this paper focuses on a comprehensive empirical study of cross-chain bridge security incidents. Specifically, we obtain cross-chain bridge security reports covering 49 cross-chain bridge security incidents from June 2021 to September 2024 from well-known security organizations such as SlowMist [31], Rekt [9] and Certik [6]. Subsequently, we construct a dataset containing 203 attack transactions and 40,000 normal attack by heuristic methods and propose a cross-chain attack detection method based on cross-chain transaction execution graphs (xTEGs). In experiments, we first evaluate the effectiveness and efficiency of BridgeGuard, then compare it with existing state-of-the-art tools. Our work contributes primarily in three aspects:

- **Comprehensive analysis:** To the best of our knowledge, this paper is the *first* to conduct an in-depth analysis on the issue of cross-chain bridge attacks from the perspective of on-chain transactions, and collect the most comprehensive dataset of cross-chain attacks. Specifically, 49 cross-chain bridge attacks that occurred between June 2021 and September 2024 are investigated.

- **Tool design:** Based on our empirical findings, we develop a tool named BridgeGuard[3] for detecting cross-chain attack transactions. BridgeGuard integrates graph representation and network motif techniques to extract the global and local features of transactions as the basic of detection.

- **Experimental evaluation:** BridgeGuard's recall is 36.32% higher than that of state-of-the-art tools, and its final transactions per second (TPS) reached 65 transactions. In addition, BridgeGuard can detect attack transactions that are not disclosed in security reports.

## 2 Understanding Bridge Attacks in Real-World

### 2.1 Cross-chain Bridge Business Logic

Cross-chain bridges are decentralized applications that serve as channels connecting different blockchain networks, enabling the transfer and exchange of assets and data across different chains [23]. Implementation of cross-chain bridges can be achieved through methods such as atomic swaps [13], relay chains [17], sidechains [30], etc. Typically, a normal and complete cross-chain bridge business workflow will have three phases: source chain, off-chain, and target chain. As shown in Fig. 1, the complete cross-chain flow is demonstrated.

- **On source Chain:** (1) The user initiates a **deposit transaction** request on the source chain to the router smart contract of the cross-chain bridge. (2) The router contract forwards the request to the corresponding token contract. (3) The token contract lock the asset in the vault and generate a lock event. (4) The Router contract verifies the authenticity of the locking event, and then generates the deposit event.

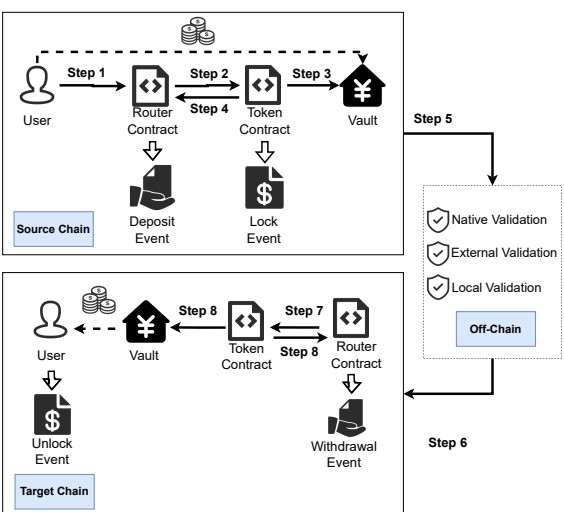

**Figure 1: Typical cross-chain bridge procedures.**

- **Off chain:** (5) The source chain message is passed down the chain. (6) The off-chain verifies that the source chain information is reliable and then passes the information to the target chain. The off chain verification methods include native verification, local verification and external verification.

- **On target chain:** (7) The router contract forwards the verified request to the token contract. (8) The token contract initiates a **withdrawal transaction**, which transfers or mints funds from the vault to the user and generates an unlock event. (9) The router contract receives the unlock event and generates the corresponding withdrawal event.

The cross-chain transaction process of cross-chain bridges typically involves communication and asset transfer between multiple blockchains, offering users the convenience of cross-chain asset exchange. However, this process also introduces complex security challenges, making cross-chain bridges a target for attackers.

### 2.2 Analyzing Bridge Attacks

*2.2.1 Data Collection and Statistic.* To summarize the attacks on cross-chain bridges, we first collect real cross-chain bridge attack incidents with two main sources:

**Academic Sok papers on cross-chain bridge attacks.** Zhang *et al.* [43] counted 31 cross-chain bridge attack cases that occurred from July 2021 to July 2023. In addition, Notland *et al.* [22] counted 34 cross-chain bridge attacks that occurred from 2021 to 2023. However, these two papers do not cover or summarize the new attack incidents and patterns that occurred after 2023.

**Attack incident summarized by security companies.**

- Slowmist[4] provides a chronological list of blockchain attack incidents, with a total of 1,497 cases recorded so far, including 42 incidents related to cross-chain bridges.

- Rekt News[5] ranks attack cases by the amount of loss incurred. Currently, it has recorded 92 cases.

---

[3]https://anonymous.4open.science/r/BridgeGuard-220E/

[4]https://hacked.slowmist.io/en/
[5]https://rekt.news/zh/leaderboard/

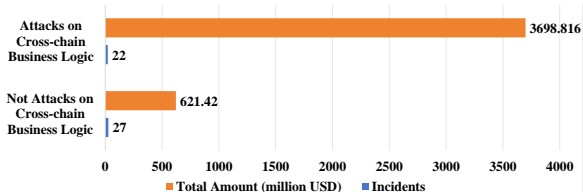

Figure 2: Statistic results of bridge attack incidents.

- ChainSec[6] archives attacks related to decentralized finance (DeFi), categorized by different blockchains and years, containing 148 DeFi attack incidents.

Finally, we collect **49 cross-chain bridge attack incidents** that occurred between June 2021 and September 2024, which is the largest academic dataset as far as we know. The comprehensive list of cross-chain bridge attack incidents is shown in the Appendix A.3, which includes details such as the attacked cross-chain bridges, attack date, the amount of losses, information source, attack stage of cross-chain, and reasons.

Based on the cross-chain business logic introduced in Section 2.1, we first categorize the collected attack incidents as either against or not against cross-chain business logic. First, we investigate the incidents not against cross-chain business logic. These incidents include private key leaks, flash loans, rug pulls, front-end hacking, etc. See more details in Appendix A.3. Then, we analyze the number of incidents and the corresponding financial losses. As shown in Fig. 2, out of the 49 incidents we collected, 27 did not against cross-chain business logic. However, the financial losses caused by attacks against cross-chain business logic were nearly six times greater than those from non-cross-chain business logic attacks.

## 2.3 Attack on Cross-chain Bridge Business logic

Attacks against cross-chain business logic cause significantly more damage than those that do not. Thus, we focus our analysis of attacks against cross-chain bridging business logic.

*2.3.1 Attack on Source Chain (denoted as $\mathcal{A}_{src}$).* This attack happens in the source chain where the deposit transaction occurs.
**Attacks on Token Contracts.** Token contracts on the source chain, whose main function is to lock tokens and generate token-locking proofs. In this type of attack, the hacker first locks a small number of tokens or none at all. Then, by triggering a cross-chain business vulnerability in the token contract, the hacker generates proof of locking beyond the amount locked in the first step, in order to spoof subsequent cross-chain business validation. Here we provide an example of an attack incident that occurred on the Meter.io bridge on February 5, 2022, resulting in an estimated loss of $4.2 million. The Meter.io bridge offers two deposit methods, `deposit()` and `depositETH()`. However, the `deposit()` function failed to prevent the deposit of ERC20 tokens and did not correctly execute the burning or locking logic for cross-chain deposits. As shown in Fig. 3, this allowed the hacker to simulate a deposit action on the source chain by using the `deposit()` function.
**Attacks on Router Contracts.** Attacks against router contracts are built on the basis that there is an existing token lock, but the business logic of the router contract is faulty, thus generating a fake

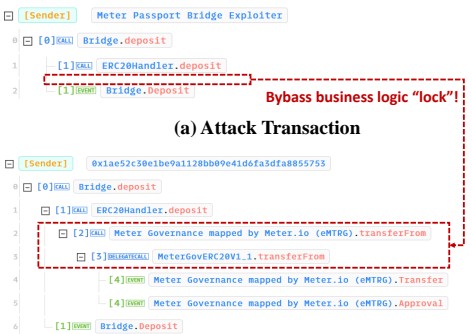

(a) Attack Transaction

(b) Normal Transaction

Figure 3: Traces of the attack and normal transactions of Meter.io Bridge

deposit event. Take ThorChain #1 as an example. In the ThorChain #1 incident[7], the attacker performed a token lock, but the token's ERC20 token symbol is "ETH". However, there is a logic error in the router contract that recognizes the tokens that are topped up as genuine Ether ETH.

> **Finding 1:** In attacks against the source chain, attack transactions often exhibit abnormal function call chains and unexpected triggering of specific contract events.

*2.3.2 Attack Off Chain.* Most off-chain attacks against cross-chain Bridges are aimed at external authentication. This may be because the bridges will choose an external verification mechanism to achieve fast multi-chain adaptation. There exists an impossible triangle for cross-chain interoperability, which means that any cross-chain scheme design cannot balance scalability, no need for trust, and easy adaptation [3]. While external validation enables fast multi-chain adaptation, it introduces new trust assumptions. Therefore, the external verification approach is one of the more fragile of all cross-chain mechanisms.

In Section 2.1, we mention that once a valid deposit event is generated, an off-chain repeater monitors and acquires the event. The repeater then passes this information to the target chain. However, if the off-chain repeater is in the hands of an attacker, then the attacker can pass the information directly to the target chain without having to make a deposit on the source chain. Using the Levyathan incident[8] as an example, we explain how this type of event is generated. The Levyathan project's tokens have a `mint()` function that allows its owner contract, `MasterChef`, to mint new tokens. While TimeLock is the owner of `MasterChef`, the `Timelock` itself should have only been operated by a multi-signature contract; however, the hacker took ownership of the `Timelock`.

> **Finding 2:** Most off-chain attacks target cross-chain bridges that use external authentication mechanisms and do not construct malicious cross-chain transactions on chain.

*2.3.3 Attack on Target Chain (denoted as $\mathcal{A}_{tgt}$).* This attack happens on the target chain where the withdrawal transaction occurs.

---

[6]https://chainsec.io/defi-hacks/
[7]https://hacked.slowmist.io/zh/?c=Bridge
[8]https://rekt.news/levyathan-rekt/

Figure 4: Traces of the attack and normal transactions in the pNetwork Bridge incident.

Attacks on the target chain mainly target router contracts, since the withdrawal operation usually has to be initiated by a router contract. Once the cross-chain business logic of the router contract is faulty, it can result in the hacking of funds. We use ChainSwap[9] as an example to explain how such events arise. In the router contract on the target chain, there is a receive function for verifying the existence of a lock event on the source chain. However, the receive function does not check the legitimacy of the incoming signer. As a result, an attacker can fool the ChainSwap's router contract on the target chain by simply generating a random address and generating a corresponding signature. In the attack against the target chain, we find some characteristic patterns of the attack transactions, such as the attacker creating an attack contract and then self-destructing, which directly triggers the router contract mint of the target chain.

> **Finding 3:** In attacks against the target chain, attack transactions often exhibit similar characteristic patterns as those attacks against the source chain.

## 2.4 Discussion of Attack Transactions

We present a comparison of normal transactions and attack transactions on cross-chain bridges from two perspectives: trace and call chain. This provides a more intuitive understanding of their differences and offers insights for the subsequent design of a cross-chain transaction detection tool.

**Transaction patterns of $\mathcal{A}_{src}$.** We take the Meter.io Bridge incident as an example of $\mathcal{A}_{src}$. Fig. 3 shows the trace comparison of the attack transaction *0x2d39* and a normal transaction *0x0ad55* on Meter.io Bridge. It is illustrated that the cross-chain business logic is not executed correctly, allowing the hacker to bypass the deposit logic. To further observe the execution process, we visualize the call chain from the trace data in Fig. 3, focusing on each CALL or DELEGATECALL, with the caller as the starting point and the callee as the endpoint. The call chains for the attack and normal transactions are presented in Fig. 5. It is evident that the hacker's call chain of the attack transaction is shorter, i.e., lacks the transfer of ERC20

[9]https://rekt.news/chainswap-rekt/

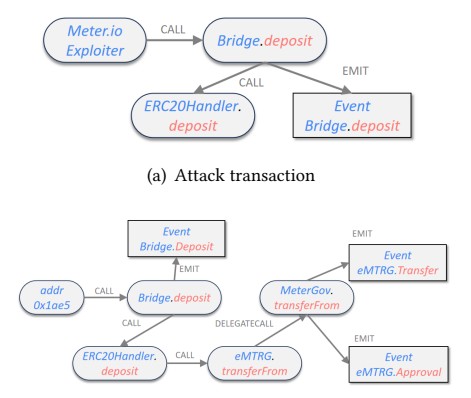

(a) Attack transaction

(b) Normal transaction

Figure 5: The call chain obtained from the traces of transactions of Meter.io Bridge

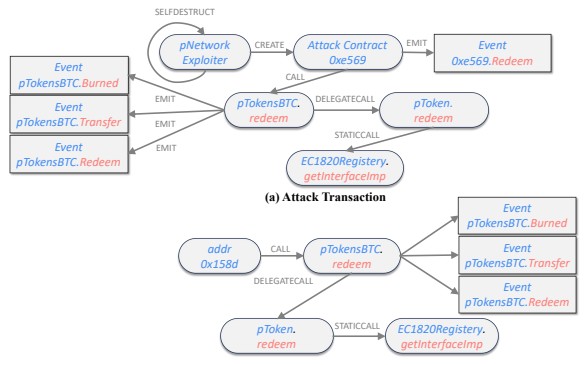

(a) Attack Transaction

(b) Normal Transaction

Figure 6: The call chain obtained from the traces of transactions of pNetwork Bridge

tokens, indicating successful bypassing of the deposit business logic on source chain.

**Transaction patterns of $\mathcal{A}_{tgt}$.** We take the pNetwork Bridge incident as an example of $\mathcal{A}_{tgt}$. In the withdrawal process on the target chain, pNetwork failed to correctly interpret the withdrawal event, resulting in the initiator of the withdrawal event being the hacker's address rather than the cross-chain bridge address. Fig. 4 displays the trace data of both the attack transaction *0x0eb55* and a normal transaction *0xeda1* on the pNetwork. Fig. 6 illustrates the call chain. It can be observed that the attacker first created the attack contract, and the lack of validation by the cross-chain bridge on the legitimacy of the initiator resulted in the attacker successfully initiating the withdrawal event using the attack contract. Subsequently, upon completion of the attack, the attacker invoked the selfdestruct() function to destroy the contract.

The existing attack detection method, XScope, requires a security pattern check of transaction pairs in the complete cross-chain process. However, we found that in many cross-chain bridge attacks, attackers may exploit off-chain verification vulnerabilities or manipulate verification mechanisms, resulting in transactions of $\mathcal{A}_{src}$ or $\mathcal{A}_{tgt}$ that lack corresponding deposit or withdrawal transactions on the target or source chain, respectively. In the dataset we collected, 65.7% of attack transactions can not find corresponding

deposit or withdrawal transactions on the target or source chain. This limitation affects the detection capability of XScope.

Even when a transaction on the source chain is identified as an attack transaction, it is often difficult to establish a clear link with the withdrawal transaction on the target chain. The complexity of cross-chain business logic makes the superficial information of attack transactions on the source chain appear completely normal compared to legitimate operations on the target chain. For these attacks, where it is challenging to find links to deposit or withdrawal transactions, analyzing the execution flow of individual transactions provides an effective solution.

> **Finding 4:** Single attack transactions on the cross-chain bridge, both on the source and target chain (i.e., $\mathcal{A}_{src}$ and $\mathcal{A}_{tgt}$), exhibit different transaction patterns in their transaction structures compared to normal transactions.

## 3 BridgeGuard: Detecting Cross-chain Attacks Transactions

### 3.1 Challenges and Solutions

Due to the complexity and significance of cross-chain bridges, efficiently detecting cross-chain attack transactions is not a trivial task. Although significant efforts have been made in the field of DeFi smart contracts [7, 19], such as reentrancy attacks [18, 28], honeypot attacks [35], and flash loan attacks [27], the rules designed in these works do not take into account the potential defects of cross-chain bridge DApps. According to the analysis and findings in Section 2, we summarize the challenges (C) faced by our attacks transactions detection tool for cross-chain bridges, BridgeGuard, and give our proposed solutions.

**C1: Expressing the execution process of cross-chain bridge transactions in a generic manner.** The operation of cross-chain bridges involves complex interactions between multiple on-chain and off-chain components. Therefore, we need a universal and precise method to represent the execution process of these transactions. Additionally, the execution of cross-chain transactions involves various associated relationships, including asset transfers, cross-chain verifications, event triggering, etc., which are difficult to be effectively represented in existing works (such as XScope [42]). Since cross-chain transactions may involve calls and interactions between multiple contracts, a more flexible and comprehensive approach is needed to capture and represent these complex associated patterns. This approach not only needs to consider the internal logical relationships of transactions, but also needs to span across different chains to fully understand the execution process of cross-chain transactions.

**C2: Identifying the differences in patterns between cross-chain attacks and normal transactions.** Based on our empirical research analysis, cross-chain bridge attack transactions may occur on either the source chain or the target chain (i.e., $\mathcal{A}_{src}$, $\mathcal{A}_{tgt}$). According to Findings 1 and 3, cross-chain attack transactions have characteristic patterns, such as abnormal function call chains and unexpected triggers of specific contract events. Therefore, we need a method to accurately identify these pattern differences to distinguish between normal transactions and potential attack behaviors. To achieve this goal, we characterize the features of the Cross-chain

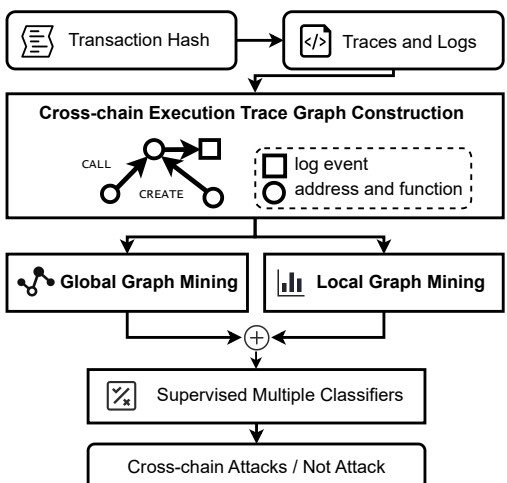

**Figure 7: The workflow of BridgeGuard.**

Transaction Execution Graph (xTEG) at both coarse and fine levels to comprehensively express transaction patterns.

To address C1, we propose the modeling method of *Cross-chain Transaction Execution Graph (xTEG)* (see Section 3.2.1). Through this method, the relationships between each invocation and the called contracts and functions in the transactions can be clearly expressed. To address this C2, we perform global graph mining on xTEGs by mapping high-dimensional data to low-dimensional vectors, and perform local graph mining focuses on identifying recurring substructures in xTEGs, which represent specific contract execution patterns.

### 3.2 BridgeGuard Overview

As shown in Fig. 7, BridgeGuard detects cross-chain business logic attacks that occur on both the source and target chains. Bridge-Guard starts from the cross-chain bridge attack event and obtains the log information and execution information of the attack transaction. BridgeGuard uses the tool BlockchainSpider [38] to obtain transaction-related data, including logs, traces, token transfers, and other information. Then, this information into a cross-chain transaction execution graph. After that, BridgeGuard conducts global graph mining and local graph mining of xTEG. Finally, BridgeGuard conducts attack detection based on supervised multiple classifiers. The pseudocode of BridgeGuard is illustrated in Algorithm 1.

*3.2.1 Cross-chain Transaction Execution Graphs Construction.* We construct cross-chain transaction execution graphs (xTEGs) to represent deposit or withdrawal operations on cross-chain business processes by taking the execution and log information as inputs. Specifically, the graph is defined as follows.

DEFINITION (Cross-chain Transaction Execution Graph, xTEG): *For a given transaction, the execution trace graph (xTEG) can be represented as a directed graph $xTEG = (V, E)$, where $V$ denotes the set of vertices*

$$v \in V = \begin{cases} \text{EOA address,} \\ \text{Contract address and function,} \\ \text{Log event,} \end{cases}$$

---

**Algorithm 1** BridgeGuard

---

**Input**: Transaction hash $tx$

**Output**: Transaction category $c$

1: $trace \leftarrow$ GETTRACE($tx$)
2: $log \leftarrow$ GETLOG($tx$)
3: $xTEG \leftarrow$ BUILDXTEG($trace$)
4: $global\_feature \leftarrow$ CONCAT(GRAPH2VEC($TEG$), STATISTIC($xTEG$), $log$)
5: $local\_feature \leftarrow$ MOTIF_COUNT($xTEG$)
6: $features \leftarrow$ CONCAT($global\_feature$, $local\_feature$)
7: $c \leftarrow$ CLASSIFIER($features$)
8: **return** $c$

---

and the set of edges E represents various types of operations:

$$e \in E = \begin{cases} \text{CALL, STATICCALL, DELEGATECALL, CALLCODE,} \\ \text{CREATE, CREATE2,} \\ \text{SELFDESTRUCT, EMIT} \end{cases}$$

*3.2.2 Global Graph Mining of xTEG .* Graph embedding techniques can effectively transform high-dimensional discrete graph data into low-dimensional continuous vector spaces, maximally preserving the structural properties of the graph [40]. Therefore, We employ a graph embedding method Graph2vec [20] to learn global features in xTEG with the following key advantages: unsupervised representation learning that captures structural equivalence, i.e., structurally similar graphs can produce similar embeddings. Graph2vec algorithm extends the concept of word embedding from the Doc2vec algorithm [14] in natural language processing to graph embedding. It treats the entire graph as a document and considers each vertex's rooted subgraph (i.e., neighborhood) as the words in the document. The basic idea of using Graph2vec for xTEG graph mining is as follows: for each vertex in the xTEG, it first generates rooted subgraphs using the Weisfeiler-Lehman kernel (WL kernel) [29] and assigns unique labels to these subgraphs. Then, it treats the collection of all rooted subgraphs around each vertex as its vocabulary. Finally, it employs the Skip-gram optimization model [11] from Doc2vec to learn vector representations for each xTEG in the dataset.

In addition, besides structural features, basic statistical features of the graph also differ between attack transactions and normal transactions. Therefore, BridgeGuard additionally computes four global graph metrics: the number of nodes $|V|$, the number of edges $|E|$, the number of logs, and network density $D = \frac{2|E|}{|V|(|V|-1)}$. In addition, we mark each transaction as a deposit or withdrawal by identifying functions in the log. To this end, BridgeGuard obtains the global feature $F_{glo} \in R^{21}$ of xTEG.

*3.2.3 Local Graph Mining of xTEG .* In BridgeGuard's task, it is insufficient to merely characterize the contract execution process globally, as this may only distinguish between attacking and non-attacking transactions. BridgeGuard needs to further distinguish which type of defect causes attacking transactions, therefore, subsequently conducts local graph mining on xTEGs to achieve a more detailed characterization of contract execution patterns.

Network motifs are recurring subgraphs in a network, whose occurrences in complex networks are significantly higher than

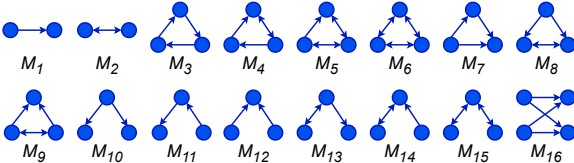

**Figure 8: Network motifs.**

in random networks [2]. They serve as the fundamental building blocks of networks and are effective tools for revealing higher-order network structures. Inspired by Benson *et al.* [5], we consider motifs to be a pattern of edges on a small number of nodes, as shown in Fig. 8. For each transaction's xTEG, BridgeGuard calculates the frequency of occurrence of these 16 motifs. Specifically, BridgeGuard calculates the directed motif $M_1$-$M_{16}$ by subgraph matrix computation [5, 38]. Then, BridgeGuard outputs a 16-dimensional localized feature vector, where the frequency of occurrence of the $i$-th motif is used as the $i$th-dimensional feature. To this end, BridgeGuard obtains the global feature $F_{loc} \in R^{16}$ of xTEG.

We observe that cross-chain transactions exhibit distinct features in global and local levels. Those two feature vectors are combined by concatenation, i.e., $F = F_{glo}||F_{loc}$, to get a more precise and general characterization, $F \in R^{37}$.

## 3.3 Experimental Setup

In this part, we perform experiments to demonstrate the effectiveness of the proposed tool, BridgeGuard, in protesting cross-chain bridges against attacks.

**Research questions.** In particular, we aim to answer the following research questions (RQs):

- RQ1: How effective and efficient is BridgeGurad in detecting the attack transactions of cross-chain bridge incidents?
- RQ2: How do existing attack transaction detection tools for blockchain perform in cross-chain bridges?
- RQ3: Can BridgeGurad find new cross-chain attack transactions?

**Dataset.** Based on the data collection method introduced in Section 2.2, we collect 49 incidents occurred on cross-chain bridges from June 2021 and September 2024. Given that the number of normal transactions on the chain far exceeds the number of attack transactions, in order to make the distribution of the evaluation dataset as close as possible to the actual situation, we mix 203 attack transactions into 40,000 normal transactions at a rate of 5%. In the supervised tasks, we divide the dataset into a training set and a test set with a ratio of 7:3. Besides, we conduct ten repeated experiments to obtain averaged results.

**Evaluation Metrics.** We use precision, recall, F1 score, and support to demonstrate performance. We first obtain true positives (TP), false positives (FN) and false negatives (FN).

## 3.4 RQ1: Effectiveness and Efficiency of BridgeGuard

To answer RQ1, first, we run BridgeGuard on the dataset with the supervised setting. Specifically, we obtain features $F$ including global and local features of xTEGs for each transaction. Using the 36-dimensional features as input, we utilized various supervised classifiers are utilized as classifiers, including Decision Tree [32],

**Table 1: Results of BridgeGuard under different classifiers.**

| Methods | Precision (%) | Recall (%) | F1-score (%) |
|---|---|---|---|
| BridgeGuard$_{DT}$ | 92.00 | 81.25 | 86.63 |
| BridgeGuard$_{XGBoost}$ | 80.90 | 80.00 | 80.45 |
| BridgeGuard$_{MLP}$ | 71.00 | 70.00 | 70.50 |
| BridgeGuard$_{KNN}$ | **92.00** | **86.50** | **89.16** |

**Table 2: Ablation experiments of BridgeGuard.**

| Features | Precision (%) | Recall (%) | F1-score (%) |
|---|---|---|---|
| BridgeGuard | 92.00 | 86.50 | 89.16 |
| BridgeGuard (w/o $F_{glo}$) | 88.00 | 81.00 | 84.35 |
| BridgeGuard (w/o $F_{loc}$) | 82.00 | 85.00 | 83.47 |

eXtreme Gradient Boosting (XGBoost) [8], Multilayer Perceptron (MLP) [26], and K-Nearest Neighbor (KNN) [25].

Table 1 shows the classification results of attack transactions detection on cross-chain bridges. The attack transactions are treated as the positive sample and the other defects are treated as the negative sample. The experimental results demonstrate that the proposed features achieve an F1 score of over 70% across several classifiers, though there are significant differences in performance between them. Notably, BridgeGuard$_{KNN}$ performs the best with an F1 score of 90.25%, likely due to its adaptability to nonlinear decision boundaries. Therefore, BridgeGuard$_{KNN}$ is selected as the primary classifier in subsequent experiments.

To further validate the contribution of each feature of the proposed BridgeGuard, we conduct an ablation study as follows. We separately remove the global features (i.e. w/o $F_{glo}$) or remove the local features (i.e., w/o $F_{loc}$). The results are as shown in Table 2. Overall, recall is higher when using global features, while precision is higher when using local features. This may be due to the fact that global features are more inclined to capture the overall structure and major patterns, while local features are more focused on local details and specific structures. We can see that using only global features and using only local features resulted in a decrease in precision of 10% and 4%, respectively. Thus, the combination of global and local graph mining enables us to better capture the characteristics of transaction attacks, resulting in better results.

To evaluate the efficiency of BridgeGuard in practical detection, we conduct experiments to measure the time taken for its identification process. These experimental results are crucial for determining the usability and scalability of BridgeGuard in real-world environments. We record the detection time for different steps in BridgeGuard and list the results in Table 3.

**Table 3: The time consumption of BridgeGuard.**

| Step | Avg. Time ($second^{-3}$) |
|---|---|
| xTEG Construction for Transactions | 0.253 |
| Global Graph Mining | 0.332 |
| Local Graph Mining | 14.6 |
| Attack Detection Classifier | 0.027 |
| Total | **15.212** |

As shown in Table 3, BridgeGuard's final transactions per second (TPS) reached 65 transactions (i.e., $\frac{1000}{15.212}$), whereas the average TPS of Ethereum is 12.4 [1]. Therefore, by pre-executing transactions in the pending transaction pool, BridgeGuard has the capability to

**Table 4: Results of different tools in detecting cross-chain attack transactions**

| Tools | Transactions | Precision (%) | Recall(%) | F1-score(%) |
|---|---|---|---|---|
| XScope | Attack ($\mathcal{A}_{src}$, $\mathcal{A}_{tgt}$) | 100.00 | 43.68 | 60.80 |
| | Normal | 100.00 | 100.00 | 100.00 |
| DeFiScanner | Attack ($\mathcal{A}_{src}$) | 0.00 | 0.00 | 0.00 |
| | Attack ($\mathcal{A}_{tgt}$) | 0.00 | 0.00 | 0.00 |
| | Normal | 98.00 | 100.00 | 99.00 |
| BridgeGuard | Attack ($\mathcal{A}_{src}$) | 86.00 | **66.00** | 74.68 |
| | Attack ($\mathcal{A}_{tgt}$) | 90.00 | **94.00** | 91.96 |
| | Normal | 100.00 | 100.00 | 100.00 |

uncover such malicious behavior before the attack transactions are recorded on the blockchain. This efficient speed not only enhances the detection rate of malicious transactions, but also allows for timely defensive measures to mitigate potential losses. We notice that the most time-consuming part mainly lies in the local graph mining step. This is because calculations need to be performed for each network motif (a total of 16 considered in BridgeGuard), which is equivalent to traversing the entire graph multiple times. This highly computationally intensive process requires a significant amount of computational resources and time.

## 3.5   RQ2: Comparison with Existing tools

To address RQ2, we compare the performance of the state-of-the-art methods in detecting attack transactions. The methods included in the comparison are:

- XScope [42] proposes security facts and inference rules for cross-chain bridges, and then designs security properties and patterns to detect cross-chain attacks from normal transactions.
- DeFiScanner [36] focuses on detecting smart contract vulnerabilities on Ethereum from a transactional perspective. DeFiScanner employs a neural network that can detect transactions with different categories.

Table 4 presents the performance of different models in detecting cross-chain attack transactions. We observe that XScope performs exceptionally well in detecting normal transactions. However, when it comes to detecting attack transactions, the recall was only 43.68%, suggesting that XScope has a high false negative rate in detecting attack transactions. Similarly, DeFiScanner shows strong performance in detecting normal transactions (F1-score=99%), but in terms of detecting attack transactions, whether for deposit or withdrawal attacks, all metrics were zero, indicating that the tool completely failed to identify any attack transactions. In contrast, BridgeGuard not only effectively identified the majority of attack transactions (with a recall of 80%), but also demonstrated high precision, meaning that most transactions flagged as attacks were indeed genuine attack transactions.

In summary, BridgeGuard outperforms both XScope and DeFiScanner in detecting cross-chain bridge attack transactions, especially in identifying withdrawal attack transactions where BridgeGuard nearly achieves optimal performance. The recall of BridgeGuard is 42.5% higher than the Xscope tool. In contrast, XScope exhibits a high false negative rate in attack detection due to its reliance on predefined security patterns, which limits its ability to adapt to emerging attack patterns, making it easier for attackers

**Table 5: Newly detected attack transactions by BridgeGuard**

| System | Newly Detected Attack Transactions |
|---|---|
| Thorchain #1 | 0x99f95561c60471f1a07a8dec48d8d4f1f26cf82658d2c11645c515ee57c052b6 |
| | 0x1522b5a8e1256b605a987e997b295fae073ceab59895eec4b1f9eb3e22a366ca |
| pNetwork | 0x975cbc1c5f9718e1aaf41288664bc99a78952d62593487baac979f3741d81e94 |
| | 0x72beef34380fa2cf96f1320f6b3cb921f9ad371970a38fed8cbde0925cef6914 |

to evade detection. DeFiScanner, on the other hand, is almost ineffective in detecting cross-chain bridge attacks, as it is designed for general DApp attack detection and does not account for the specific business logic of cross-chain bridges (as discussed in Section 2). Therefore, BridgeGuard stands out as the most reliable option, maintaining high precision while also delivering superior recall performance.

## 3.6 RQ3: Finding New Attack Transaction

To answer whether BridgeGuard can identify new attack transactions that were previously undetected by other tools. By analyzing the false positives generated by the BridgeGuard algorithm, we successfully discover attack transactions that existing tools failed to detect. Table 5 presents the attack transactions that are newly discovered using our new tool.

- **Attack transactions in Thorchain #1 incident**: Two new attack transactions, 0x99f and 0x152 are detected. The sender of these transactions is the same as that of the reported attack transaction[10]. Based on the findings of Su et al. [34], transactions initiated by the attacker are highly likely to be attack transactions as well. We also examine the behavior of these transactions, and find that the traces and triggered functions exhibited similar patterns to the known attack transaction.
- **Attack transactions in pNetwork incident**: We also detect two new attack transactions, 0x72b and 0x975. Both of these transactions were initiated by the attacker but are not included in the security report.

The results of this study demonstrate that our approach offers significant advantages in detecting cross-chain bridge attack transactions, particularly for newly identified attacks that were previously undetected by other tools. These newly detected attack transactions provide critical reference points for future security measures and help researchers and developers gain a better understanding of potential security threats and how to mitigate them.

## 4 Related Work

### 4.1 Security Analysis of Cross-chain Bridges

Lee et al. [16] elucidated several cross-chain bridging attacks and proposed mitigations for most of them. Notland et al. [22] analyzed 34 cross-chain bridge security incidents, identifying 8 categories of critical vulnerabilities and proposing 11 mitigations. However, these studies are still insufficient in the systematic and comprehensive analysis of attacks, and may not cover all potential attack vectors. Belchior et al. [4] proposed the Hephaestus model which provides a new way of modeling the complexity of cross-chain applications, but its applicability in real-world environments has yet to be verified. Zhang et al. [42] discovered three types of vulnerabilities in

cross-chain bridges and proposed the Xscope monitoring tool, but its validity and extensibility still require However, its effectiveness and scalability still need to be further studied. Therefore, future research should focus on integrating the existing results and conducting more systematic empirical analysis to improve the security and reliability of cross-chain bridges.

### 4.2 Detection for DeFi Attacks

Research on DeFi attacks can be divided into two types: detecting from a contract perspective and detecting from a transaction perspective. From the perspective of contracts, Rodler et al. [28] mainly used the execution flow analysis method to detect re-entry vulnerabilities in contracts. And Chen et al. [7] developed a tool that can detect contract security online and expand to custom vulnerabilities. From the perspective of transactions, Zhou et al. [48] conducted a large-scale measurement and analysis of Ethereum transaction logs for the first time and discovered some new types of attacks, such as airdrop hunting. However, Su et al. [34] focused on existing attack cases and proposed the tool DEFIER to automatically investigate large-scale attack events. Zhang et al. [44] hoped to develop a universal attack detection framework, which detects the security of transactions by modifying Geth, replaying historical transactions, and defining a series of security attributes. In addition, Zhou et al. [47] studied how to systematically measure, evaluate, and compare DeFi attack events. The paper [36] focuses on the detection of logical vulnerabilities on Ethereum. Su et al. [33] analyzed token leakage vulnerabilities by mining the relationship between users and DApps.

## 5 Conclusion and Future Work

In this paper, we conducted an in-depth study of cross-chain bridge attack incidents and proposed a detection tool for attacks targeting cross-chain business processes, called BridgeGuard. By collecting and analyzing 49 cross-chain bridge attack incidents, particularly those against cross-chain business processes, we constructed cross-chain transaction execution graphs (xTEGs) and extracted statistical and structural features. Experimental results show that BridgeGuard demonstrates excellent performance in detecting cross-chain attacks, with a recall rate 42.5% higher than the state-of-the-art tools and the ability to identify newly discovered attack transactions. We believe that the introduction of BridgeGuard provides an effective solution to enhance the security of cross-chain bridges, while also serving as an important reference for future research in cross-chain bridge security.

For future work, we plan to explore several directions. Firstly, we wish extend BridgeGuard to other types of cross-chain bridges, such as NFT bridges and governance bridges, to achieve more comprehensive cross-chain security monitoring. Additionally, we can optimize the performance of BridgeGuard, including improving detection efficiency and reducing resource consumption, to meet the requirements of real-world applications. Finally, we can explore the application of large-scale language model (LLM) in cross-chain security to improve the recognition and defense against complex attack patterns.

---

[10] 0x92b466c1908571c45b1a4e751550877a46c5f2ecbc308e01242ec6d2013ad88c

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

# A   Appendix

## A.1   Background

*A.1.1   Blockchain.* Blockchain technology was first introduced by Satoshi Nakamoto in the Bitcoin whitepaper in 2008, representing a distributed and tamper-resistant ledger system designed to address

Table 6: Cross-chain bridge attack incidents and the corresponding taxonomy.

| Incidents | Attack Date | Incident Loss ($) | Information Source | Attack Stage of Cross-chain | Reason |
|---|---|---|---|---|---|
| THORChain #2 | 2021/07/16 | 5,000,000 | Rekt News | Source Chain | Fake Lock Event |
| Qubit | 2022/01/01 | 80,000,000 | Rekt News | Source Chain | Fake Lock Event |
| Meterio | 2022/02/06 | 4,200,000 | Rekt News | Source Chain | Fake Lock Event |
| THORChain #1 | 2021/06/29 | 350,000 | Slowmist | Source Chain | Fake Deposit Event |
| THORChain #3 | 2021/07/23 | 8,000,000 | Rekt News | Source Chain | Fake Deposit Event |
| QAN Platform | 2022/10/11 | 2,000,000 | Rekt News | Source Chain | Fake Deposit Event |
| Anyswap #1 | 2021/07/10 | 7,900,000 | Rekt News | Off-chain | Verification failure |
| Levyathan | 2021/07/30 | 1,500,000 | Rekt News | Off-chain | Verification failure |
| Ronin #1 | 2022/03/29 | 625,000,000 | Rekt News | Off-chain | Verification failure |
| Rainbow(NEAR) #1 | 2022/05/02 | 0 | Notland *et al.* [21] | Off-chain | Verification failure |
| Nomad | 2022/08/01 | 190,000,000 | Rekt News | Off-chain | Verification failure |
| Binance bridge | 2022/10/08 | 566,000,000 | Rekt News | Off-chain | Verification failure |
| Poly Network #2 | 2023/07/01 | 10,200,000 | Rekt News | Off-chain | Verification failure |
| Ronin #2 | 2024/08/06 | 12,000,000 | Rekt News | Off-chain | Verification failure |
| Poly Network #1 | 2021/08/11 | 600,000,000 | Rekt News | Off-chain | Verification failure |
| ChainSwap | 2021/07/11 | 8,000,000 | Rekt News | Target Chain | Unverified withdrawal |
| pNetwork | 2021/09/20 | 13,000,000 | Medium | Target Chain | Unverified withdrawal |
| wormhole | 2022/02/03 | 320,000,000 | Rekt News | Target Chain | Unverified withdrawal |
| Ankr | 2022/12/02 | 24,000,000 | Rekt News | Target Chain | Unverified withdrawal |
| Hypr bridge | 2023/12/14 | 220,000 | Rekt News | Target Chain | Unverified withdrawal |
| X bridge | 2024/04/24 | 1,440,000 | Rekt News | Target Chain | Unverified withdrawal |
| Polygon Plasma | 2021/10/21 | 850,000,000 | Medium | Target Chain | Unverified withdrawal |
| Zapper | 2021/06/10 | 0 | Notland *et al.* [21] | Not specific to cross-chain process | Over-Authorisation |
| Anyswap #2 | 2022/01/18 | 3,000,000 | Notland *et al.* [21] | Not specific to cross-chain process | Over-Authorisation |
| Li Finance | 2022/03/20 | 600,000 | Medium | Not specific to cross-chain process | Over-Authorisation |
| Badger | 2022/12/02 | 120,000,000 | Rekt News | Not specific to cross-chain process | Over-Authorisation |
| Rubic | 2022/12/25 | 1,400,000 | Slowmist | Not specific to cross-chain process | Over-Authorisation |
| Hashflow | 2023/07/14 | 600,000 | Medium | Not specific to cross-chain process | Over-Authorisation |
| Socket tech | 2024/01/16 | 3,300,000 | Notland *et al.* [21] | Not specific to cross-chain process | Over-Authorisation |
| ALEX Lab | 2024/05/15 | 4,300,000 | Rekt | Not specific to cross-chain process | Private key leakage |
| Hector Network | 2024/01/15 | 27,000,000 | Notland *et al.* [21] | Not specific to cross-chain process | Private key leakage |
| Orbit chain | 2023/12/31 | 81,500,000 | Rekt News | Not specific to cross-chain process | Private key leakage |
| Heco bridge | 2023/11/22 | 99,100,000 | Rekt News | Not specific to cross-chain process | Private key leakage |
| pGala | 2022/11/04 | 10,800,000 | Slowmist | Not specific to cross-chain process | Private key leakage |
| Harmony | 2022/06/23 | 100000000 | Rekt News | Not specific to cross-chain process | Private key leakage |
| Marvin Inu | 2022/04/11 | 350,000 | Notland *et al.* [21] | Not specific to cross-chain process | Private key leakage |
| Allbridge | 2023/04/01 | 57,000,000 | Medium | Not specific to cross-chain process | Flash-loan |
| Zenon | 2021/11/21 | 1000000 | Rekt | Not specific to cross-chain process | Flash-loan |
| Multichain | 2023/07/06 | 126,300,000 | Rekt News | Not specific to cross-chain process | Rug-pull |
| Ordizk | 2024/03/05 | 14,000,000 | Certik | Not specific to cross-chain process | Rug-pull |
| Bondly | 2021/07/15 | 5,900,000 | Rekt News | Not specific to cross-chain process | Rug-pull |
| LayerSwap | 2024/03/20 | 100,000 | Slowmist | Not specific to cross-chain process | DNS hijacking |
| Celer Bridge | 2022/08/18 | 20,000 | Slowmist | Not specific to cross-chain process | DNS hijacking |
| EvoDeFi Bridge | 2022/03/08 | 0 | Slowmist | Not specific to cross-chain process | DNS hijacking |
| deBridge | 2022/08/06 | 0 | Notland *et al.* [21] | Not specific to cross-chain process | Phishing email |
| Rainbow(Aurora) | 2022/05/02 | 0 | Notland *et al.* [21] | Not specific to cross-chain process | False transaction |
| Rainbow(NEAR) | 2022/08/22 | 0 | Notland *et al.* [21] | Not specific to cross-chain process | Fabricated block |
| Omni Bridge | 2022/09/16 | 4,200,000 | Notland *et al.* [21] | Not specific to cross-chain process | Replay attack |
| Meson Finance | 2024/04/19 | 0 | Slowmist | Not specific to cross-chain process | Hacked twitter |

trust among multiple parties in a public ledger [46]. This technology operates on a peer-to-peer network, where each participant or node holds a copy of the entire blockchain. By leveraging cryptographic techniques, blockchain packages transactions into blocks and links them together in a chain, ensuring the security and transparency of data. Each block contains the hash value of the previous block,

providing the blockchain with immutability and enabling trusted transactions without the need for intermediaries. Blockchains can be categorized into public chains, private chains, consortium chains, and others.

The emergence of multi-chain ecosystems has made it challenging for assets and data to interoperate between different blockchain networks. Currently, the blockchain ecosystem consists of multiple chains, with records of 260 public blockchains as of March 2024, according to data from DeFiLlama[11]. However, data between different blockchain systems are not interoperable, akin to isolated islands. Therefore, DeFi bridges, as applications capable of facilitating asset circulation and information exchange between chains, can promote further development of the multi-chain ecosystem [15].

*A.1.2 Transaction and Smart Contract.* Transaction data is a type of data on the blockchain. Transactions are data structures that record cryptocurrency information on the blockchain, which can be messages sent to smart contracts or simple token transfers to blockchain users. Transactions are fundamental units of activity on the blockchain, representing a modification to the blockchain's state. In blockchains like Bitcoin and Ethereum, transactions typically include sender addresses, recipient addresses, amounts of assets transferred, transaction fees, and other information. Once a transaction is created, it is broadcasted to all nodes on the network, undergoes validation, and is included in a new block. Transactions on the blockchain are irreversible; once confirmed, they are permanently recorded on the blockchain.

Smart contracts are another data type in the blockchain domain, initially invented in Ethereum. Smart contracts are self-executing contracts that run on the blockchain, where the terms and conditions are programmatically defined and executed by the blockchain network [45]. Smart contracts are typically written in programming languages like Solidity and deployed onto the blockchain for execution. Once deployed on the blockchain, smart contracts become immutable. They automatically execute once their predefined conditions are met, without the need for third-party intervention.

## A.2 Cross-chain Bridge Attack Incidents List

The comprehensive list of cross-chain bridge attack incidents is shown in Table 6, which includes details such as the attacked cross-chain bridges, attack date, the amount of losses, information source, attack stage of cross-chain, and reasons.

## A.3 Attack that not against cross-chain business logic

These incidents can be concluded as these categories:

- **Private Key Leakage [12].** For an EOA account, the account consists of a public and private key cryptographic pair. Its role is to prove that the transaction was actually signed by the sender and to prevent forgery. For individuals, the private key is the key used to sign transactions, so it is used to safeguard the user's management of the funds associated with the account. If a user compromises their account private key, a hacker will be able to silky-smoothly transfer any asset within their account.

---
[11]https://defillama.com/chains

- **Over-Authorisation [24].** A DeFi app obtaining authorisation from users is likely to be at risk of over-authorisation. Authorisation is essentially an on-chain transaction that requires the user to pay gas fee, and in order to avoid repeated authorisations by the user, the developer of a DeFi app will usually set the maximum number of tokens to be authorised to the smart contract by default. However, such a process also obviously exposes the risk, if the smart contract has a loophole or the contract administrator is evil, then the user's tokens will be at risk of loss, which is the problem of over-authorisation of the Dapp.
- **Others.** Other attacks that are not specific to cross-chain bridges include flash-loan [27], rug pull [49], front-end hacking [10], etc.

## B Discussion

**Internal Validity.** BridgeGuard focuses on attacks caused by on-chain contract defects, while attacks caused by off-chain components are not considered in this paper. Additionally, although BridgeGuard currently supports the detection of four types of on-chain contract defects, its method is based on xTEG, which allows for the detection of additional types of defects. Specifically, in global graph mining, the training parameters of Graph2vec can be adjusted as needed, such as setting a larger embedding dimension to retain more information. In local graph mining, new computing modules can be added based on the substructure features of newly identified defect types. Finally, it is worth noting that BridgeGuard primarily targets cross-chain bridges for fungible asset transfers, and other types of bridges such as Non-Fungible Token (NFT) bridges, governance bridges, ENS bridges, etc., are out of scope. However, our framework can easily be extended to other types of property transfers, as these transactions can also construct xTEGs for detection.

**External Validity.** In our empirical study, relying on manual labor during the data collection and organization process could introduce human errors. To mitigate this dependence, we ensure that each event was reviewed by at least two paper authors. Additionally, our dataset primarily originates from four public resources (Slowmist, Rekt and ChainSec) and two academic SoK papers (Zhang *et al.* [43] and Notland *et al.* [21]). To the best of our knowledge, these resources constitute the most extensive accessible database of cross-chain bridge incidents. However, we cannot fully evaluate whether these sources contain biased cases, as we do not know how they collect attack events. This may lead to more attacks being overlooked. Although we cannot confirm the collection pathway for a single data source, we reduce bias in our data by integrating multiple data sources.

