# OpenReview forum: "Safeguarding Blockchain Ecosystem: Understanding and Detecting Attack Transactions on Cross-chain Bridges"
_ACM.org/TheWebConf/2025/Conference — WWW 2025 Poster_

### Official Review · Reviewer_WFBW · 2024-11-24

**Novelty:** 6
**Technical Quality:** 7

**Review:**

This paper addresses the problem of cross-chain bridge attack transactions and proposes a novel detection method called BridgeGuard. It conducts an in-depth analysis of cross-chain bridge attacks from the perspective of on-chain transactions. The dataset is comprehensive, covering the largest number of cross-chain bridge attack cases to date (49 incidents). By constructing cross-chain transaction execution graphs (xTEGs) and using a two-stage detection framework, the method effectively integrates global and local graph mining to capture the execution patterns of smart contracts in cross-chain transactions.

**Questions:**

(1) The differences from related works with similar problem have not been discussed. I noticed a paper called SmartAxe that also addresses a problem similar to yours. Could you clarify what are the significant advantages or differences of your work compared to SmartAxe?

Zeqin Liao, Yuhong Nan, Henglong Liang, Sicheng Hao, Juan Zhai, Jiajing Wu, and Zibin Zheng. 2024. SmartAxe: Detecting Cross-Chain Vulnerabilities in Bridge Smart Contracts via Fine-Grained Static Analysis. Proc. ACM Softw. Eng. 1, FSE, Article 12 (July 2024), 22 pages. https://doi.org/10.1145/3643738

(2) The motivation for using xETGs is not fully clear. The introduction of xETGs is interesting, but it’s unclear whether existing tools have already explored attack transaction features from a graph perspective. I would appreciate further clarification on why xETGs were chosen.

(3) Some experimental details are unclear. The abstract mentions a recall improvement of 36.32% over SOTA methods. Please explain how this value was calculated. Section 3.2.2 mentions that global features have 21 dimensions. How many dimensions are generated by graph2vec for xETGs?

(4) Section 3.3 mentions blending 203 attack transactions into 40,000 normal transactions at a rate of 5%. Based on the numbers, this should be 0.5%—please double-check.

Please check the manuscript before submission to avoid grammatical issues.

**Reviewer Confidence:**

4: The reviewer is certain that the evaluation is correct and very familiar with the relevant literature

**Scope:**

4: The work is relevant to the Web and to the track, and is of broad interest to the community

---

### Official Review · Reviewer_mm9o · 2024-11-29

**Novelty:** 6
**Technical Quality:** 5

**Review:**

The paper presents a comprehensive study on cross-chain bridge attacks and proposes a novel detection tool called BridgeGuard. Overall, the paper presents a significant contribution to the field of blockchain security, particularly in the area of cross-chain bridges. The clarity of presentation is high, with well-structured sections explaining the background, methodology, and results.
Pros:
Comprehensive dataset: The authors collect the largest number of cross-chain bridge attack incidents to date, covering 49 attacks from June 2021 to September 2024.
Novel approach: The paper introduces BridgeGuard, a tool that uses graph representation and network motif techniques to detect cross-chain attack transactions.
Practical significance: Given the substantial financial losses in cross-chain bridge attacks, this research addresses a critical issue in blockchain security.
Clear methodology: The paper provides a detailed explanation of cross-chain bridge business logic and attack patterns.
Strong performance: BridgeGuard reportedly outperforms state-of-the-art tools with a 36.32% higher recall.

Cons:

Limited discussion on potential limitations: The paper could benefit from a more in-depth discussion of the limitations of the proposed approach.
Lack of comparison with existing solutions: While the paper mentions outperforming state-of-the-art tools, a more detailed comparison would strengthen the claims.
Ethical considerations: The paper does not discuss potential ethical implications of detecting and potentially preventing cross-chain transactions.

**Questions:**

Some questions for more clarity

1. How does BridgeGuard handle potential false positives, and what measures are in place to prevent legitimate transactions from being flagged as attacks?
2. Can you provide more details on the scalability of BridgeGuard for larger blockchain networks and higher transaction volumes?
3. How does BridgeGuard adapt to new attack patterns that may emerge in the future?
3. What are the potential limitations of using graph representation for detecting cross-chain attacks, and how might these be addressed in future work?
4. Have you considered the potential ethical implications of implementing such a detection system, particularly in terms of user privacy and transaction transparency?

**Reviewer Confidence:**

4: The reviewer is certain that the evaluation is correct and very familiar with the relevant literature

**Scope:**

4: The work is relevant to the Web and to the track, and is of broad interest to the community

---

### Official Review · Reviewer_4pyj · 2024-12-02

**Novelty:** 6
**Technical Quality:** 6

**Review:**

Pros:
This paper offers several notable strengths: it presents the largest academic dataset of cross-chain bridge attacks to date (49 incidents), introduces BridgeGuard - a novel and effective detection tool that significantly outperforms existing solutions, provides comprehensive technical analysis of attack patterns and vectors, demonstrates strong empirical results with extensive testing, and makes practical contributions by successfully detecting previously unknown attacks. The methodological approach is thorough and well-structured, combining global and local graph features in a two-stage detection framework that achieves superior recall rates compared to state-of-the-art tools.

Cons:
•	The article cites a large number of existing studies. Compared with existing studies, Bridge Guard's incremental contribution is limited and may lack outstanding innovation.
•	The comparison benchmarks are limited to XScope and DeFiScanner, both published in 2022, lacking comparisons with more cutting-edge methods to fully demonstrate its innovation.
•	The process of local graph mining for the model is difficult to meet the real-time detection requirements in high-throughput blockchain environments outside of Ethereum (e.g. Base has more than 100 real-world TPS), affecting practical applications.
•	The experiment only has a training set and a test set, without an independent validation set, so the results are not rigorous.
•	The selection of normal transactions is missing in the article, and the ‘normal transactions’ may also contain potential attacks or attempted attacks. It is also puzzling that the rate is set at 5%. This should be an imbalance problem but the author didn’t solve.

**Questions:**

Questions:

•	What’s the normal transaction’s definition and how you pick them? Why 5%?
•	How adaptable is the system to new attack patterns that may emerge?
•	How does the system handle network latency and temporary blockchain forks(reorg)?
•	Can this appiled to the bridges on Solana and Base, the most welcomed two blockchain ecosystem?
•	What are the minimum hardware requirements for running BridgeGuard with 65tps?

**Reviewer Confidence:**

3: The reviewer is confident but not certain that the evaluation is correct

**Scope:**

3: The work is somewhat relevant to the Web and to the track, and is of narrow interest to a sub-community

---

### Official Review · Reviewer_dSMe · 2024-12-03

**Novelty:** 5
**Technical Quality:** 4

**Review:**

Quality: The article presents a rigorous and comprehensive introduction to the current state of the cross-chain bridge domain, employing a systematic approach to studying cross-chain bridge attacks. The use of graph feature extraction methods and the detailed categorization of attacks are commendable. Experimental validation, including ablation experiments and performance tests, further substantiates the claims made in the paper. However, there are some shortcomings in terms of quality. Some issues are not fully discussed, such as the previously mentioned off-chain attack problem, which seems to be overlooked in the later sections on graph methods and experiments; in data description, considerations for handling few samples of detection objects could be taken into account.

Clarity: The article is generally clear in its presentation, especially in the introduction to the cross-chain bridge domain and data description. However, there are areas where the methodology and problem formulation could be more explicit. For instance, the classification of the graph problem (node, edge, or subgraph) should be clarified to aid understanding; the necessity of referencing the LocalGraph method could also be explained, as local features might be directly obtained through adjusting the sampling range in the GlobalGraph. The choice of methods and symbol representation could be made clearer.

Originality: The article introduces a novel approach to detecting cross-chain bridge attacks using graph-based methods, which is a significant contribution to the field. However, the use of graph methods in the article is quite basic, lacking specialized modeling for the specific problem of cross-chain bridges, contributing less to the field of blockchain graph research.

Significance: The research holds significant importance in the field of blockchain security, particularly in comparison to existing detection methods. However, the article should also consider more practically useful aspects of cross-chain attack prevention and warning, such as introducing temporal graphs. These considerations are not mentioned in the method selection or future work sections of the article.

Overall, the article makes a significant contribution to the field of blockchain security with its innovative approach to detecting cross-chain bridge attacks. However, addressing the mentioned cons, particularly the inclusion of off-chain attack considerations and predictive analysis, would further enhance its quality and practical significance.

**Questions:**

Could you elaborate on how your proposed method addresses or plans to address off-chain attacks in cross-chain bridges?

Could you clarify the classification of the graph problem (node, edge, or subgraph) in your methodology? How does this classification impact the effectiveness of your approach?

How does your method handle scenarios where the number of detection objects is few? What preprocessing or augmentation techniques are employed to ensure robust detection? Could you discuss any potential limitations or challenges associated with handling a small sample size and how you mitigate these?

Why did you choose not to include temporal or predictive analysis in your current study? What are the potential benefits and challenges of incorporating such analysis into your framework?

**Reviewer Confidence:**

4: The reviewer is certain that the evaluation is correct and very familiar with the relevant literature

**Scope:**

3: The work is somewhat relevant to the Web and to the track, and is of narrow interest to a sub-community

---

### Official Review · Reviewer_Y3fg · 2024-12-05

**Novelty:** 6
**Technical Quality:** 3

**Review:**

The paper presents a tool, called BridgeGuard, to detect attacks against cross-chain bridges. The tool is constructed based on the observation that cross-chain attacks exhibit different patterns than normal transactions. It represents cross-chain transactions as directed graphs, and uses graph mining techniques to identify attack patterns in cross-chain transactions. Experimental evaluation shows that BridgeGuard considerably performs better than state-of-the-art tool in its recall score metric over labeled attack dataset.

Pros:
- Modeling the cross-chain transactions as directed graphs is interesting and effective
- Good evaluation and comparison against state-of-the-art tools
- Detection of previously-unknown attacks using the tool, highlighting the value of such a tool based on behavior modeling

Cons:
- Only 22 attacks against cross-chain business logic were considered. As a result, the general applicability of the solution is questionable.
- Over-claimed comprehensiveness in comparison to past works (e.g., SoKs) as the cross-chain attacks will continue to evolve with the target blockchains.
- Empirical analysis is limited and only describes with couple of examples.

Modeling behavior of cross-chain transactions as graphs to detect attacks is interesting. The paper does a good job in discussing the approach using representative examples. The approach is also shown to be effective based on the experimental results with overall better performance in comparison to the state-of-the-art and ability to detect previously-unknown attacks.

A major weakness of the paper is with its claims that are not well-supported by the data and evidence. The tool is based on modeling just 22 attacks against business logic of the cross-chain transactions that leaves several questions unanswered. First, there is no evidence to show that it is comprehensive and generic enough to detect future attacks. Second, it still would not detect other types of cross-chain bridge attacks (27 in the experimental set). Finally, there is possibility of a knowledgeable attacker to masquerade an attack as a normal transaction by creating fake/noisy transactions, so it would be useful to provide some discussion on the robustness of the tool.

The paper also does not provide any details on how the 22 attacks against business logic were generically modeled. It only shows specific examples of couple of such attacks (Figure 5 and 6).

Please provide details on the vulnerability that was exploited in the representative examples in Figure and 4. Please increase the size of the figures for better readability.

**Questions:**

- The paper enumerates a much bigger dataset (in thousands) of attack incidents (Section 2.2), however, can you please clarify how they only resulted in 49 cross-chain bridge attack incidents (42 were summarized by just one source of Slowmist).
- Out of 49 incidents, only 22 attacks were against the business logic of cross-chain transactions. The tool is primarily based on modeling such logic. How can the tool be used, if at all, to detect other cross-chain attacks (27 in your dataset)?
- Can you provide additional details on what vulnerabilities were exploited (and how) in the examples (Figure 3 and Figure 4) presented in the paper?

**Reviewer Confidence:**

3: The reviewer is confident but not certain that the evaluation is correct

**Scope:**

3: The work is somewhat relevant to the Web and to the track, and is of narrow interest to a sub-community